# SARS-CoV-2 and Other Respiratory Viruses in Human Olfactory Pathophysiology

**DOI:** 10.3390/microorganisms12030540

**Published:** 2024-03-07

**Authors:** Serigne Fallou Wade, Abou Abdallah Malick Diouara, Babacar Ngom, Fatou Thiam, Ndongo Dia

**Affiliations:** 1École Supérieure des Sciences Agricoles et de l’Alimentation, Université Amadou Makhtar MBOW, Rue 21x20, 2ème Arrondissement, Pôle Urbain de Diamniadio, Dakar P.O. Box 45927, Senegal; babacar.ngom@uam.edu.sn; 2Groupe de Recherche Biotechnologies Appliquées & Bioprocédés environnementaux (GRBA-BE), École Supérieure Polytechnique, Université Cheikh Anta Diop, Dakar P.O. Box 5085, Senegal; fatou54.thiam@ucad.edu.sn; 3Virology Departement, Institut Pasteur de Dakar, 36, Avenue Pasteur, Dakar P.O. Box 220, Senegal; ndongo.dia@pasteur.sn

**Keywords:** respiratory viruses, anosmia, olfaction disorders, loss of smell, COVID-19

## Abstract

Acute respiratory viruses (ARVs) are the leading cause of diseases in humans worldwide. High-risk individuals, including children and the elderly, could potentially develop severe illnesses that could result in hospitalization or death in the worst case. The most common ARVs are the Human respiratory syncytial virus, Human Metapneumovirus, Human Parainfluenza Virus, rhinovirus, coronaviruses (including SARS and MERS CoV), adenoviruses, Human Bocavirus, enterovirus (-D68 and 71), and influenza viruses. The olfactory deficits due to ARV infection are a common symptom among patients. This review provides an overview of the role of SARS-CoV-2 and other common ARVs in the development of human olfactory pathophysiology. We highlight the critical need to understand the signaling underlying the olfactory dysfunction and the development of therapeutics for this wide-ranging category of AVRs to restore the altered or loss of smell in affected patients.

## 1. Introduction

Respiratory viral infections are very common and constitute a serious health concern around the world, with new infectious diseases continuing to emerge [1,2]. Such pathologies could lead to death, particularly in the elderly, and increase the expenses of the health care system worldwide. Respiratory viruses have the propensity to infect and trigger diseases through the human lower and upper respiratory tracts. We are more interested in viral upper respiratory infections (URI) as they are considered to be one of the most common causes of olfactory dysfunction, accounting for up to 45% of all cases [3,4]. Although the alteration of smell following viral URI is noticed in several cases, little treatment is currently available. The occurrence of this alteration is termed post-viral olfactory dysfunction (PVOD) [5,6,7]. The most common viruses implicated in PVOD include parainfluenza virus, rhinoviruses (RV), respiratory syncytial virus (RSV), and coronaviruses (CoV) [4,8,9,10,11]. In 1956, the RSV was found and isolated from chimpanzees and infants suffering severe lower respiratory tract illness a year later. In older children and healthy adults, the RSV causing repeated URI is common and can lead to symptomatic UR tract diseases [12]. According to Heikkinen and colleagues, 5% to 10% of URI are attributed to RSV. Studies performed in mice have shown that RSV infections are associated with damage to olfactory receptor neurons [13]. Despite these findings, the occurrence of olfactory loss associated with RSV infections seems low. It needs further investigations in broader regions of the world and during cold and warm periods to establish a clearer picture of this virus-induced olfactory dysfunction [11,14,15]. Like the RSV, the PIV was discovered in the 1950s and is associated with lower respiratory tract infections and URIs in children. Young children infected by this virus are often diagnosed with respiratory irritants, vitamin A deficiency, or malnutrition [16,17]. Although the PIV is found in respiratory secretions, the major diagnosis is observed from pulmonary secretions and confirmed by chest x-rays [18]. The RV that targets humans is considered among the most infectious agents worldwide and is associated with mild upper respiratory tract infections in people [19,20,21]. The CoV was identified in the 1960s. Little attention was given to this family of viruses until both outbreaks of the severe acute respiratory syndrome (SARS)-CoV and the Middle East respiratory syndrome (MERS)-CoV were identified in 2003 and 2012, respectively [22,23,24]. The novel coronavirus disease 2019 (COVID-19), caused by SARS-CoV-2, has spread fast all over the world [1,2]. Both MERS-CoV and SARS-CoV-2 are highly pathogenic coronaviruses and have a substantial spatial range of epidemics areas globally, but regarding MERS-CoV, the vast majority of cases are confined to the Middle East [23,24]. A key factor in the transmissibility of these viruses is the active virus replication in upper respiratory tract (URT) tissues and, therefore, its massive excretion [25,26]. Since the beginning of the pandemic, growing reports have shown the issues of partial to complete loss of smell in patients who contracted COVID-19 [27,28,29,30] and have brought new focus to PVOD. This review mainly focuses on understanding the molecular signaling underlying the olfactory pathophysiology in COVID-19-infected human patients and common viruses that induce URT infection. The importance of using such mechanisms in order to find potential targets to overcome the loss of smell will also be discussed.

## 2. Sources and Selection Criteria

The present study intends to synthesize the current knowledge regarding the relationship between the respiratory viral pathogenesis of the olfactory system and the mechanisms underlying the loss of smell in patients infected by respiratory viruses including COVID-19. We performed a literature search using mostly databases like PubMed Central (PMC), Google Scholar, and ScienceDirect to parse original articles, meta-analyses, and systematic reviews dealing with the animal and human respiratory viruses that have a negative impact on the olfactory system functionality. For our search, we used the combination of the following keywords: respiratory virus, coronavirus, rhinovirus, parainfluenza viruses and respiratory syncytial virus, anosmia, parosmia, hyposmia, olfactory epithelium, human, mouse, hamster, loss of smell, or olfactory dysfunction were considered in this review. Only the papers that met the keyword criteria listed above were considered in this work.

## 3. Olfactory Receptor and Odorant Detection

In most animals, the functional olfactory system detects and discriminates among diverse chemical stimuli. Odors are important for behaviors such as eating, mating, and avoiding dangerous smells, including smoke, leaking propane gas, and spoiled food [31,32,33]. The importance of the behaviors leads to a strong belief that the loss of olfactory function is indirectly life-threatening [34,35]. Two different olfactory systems have been developed in mammals such as rodents: the main olfactory epithelium (MOE), also called olfactory mucosa, connected to the main olfactory bulb, and the accessory system called the vomeronasal organ (VNO) connected to the accessory olfactory bulb [36,37,38,39,40]. Here, the VNO will not be discussed. The configuration of the olfactory epithelium (OE) presents unique cytological characteristics as it contains different cell types, such as the ciliated olfactory receptor neurons (ORNs), the sustentacular supporting cells, and the cells of Bowman’s glands. The olfactory mucosa hosts many different types of cells, including ORNs in the intermediate layer, the sustentacular cells on the apical and basal sides, and the sensory cilia present at the apical pole where the dendrites of olfactory neurons are extended [41,42].

A deep understanding of the molecular signaling of smelling recognition is required to understand the basis of the olfactory system and, consequently, the loss of olfactory function. Starting from the beginning of the 1990s, pioneers have developed and studied the physiology of the olfactory system based on molecular biology, biochemistry, anatomy, and bioinformatics [31,43]. At first glance, getting insight into the molecular mechanisms of the perception of odors has emerged from several disciplines such as chemistry, biology, and professional odor detectors [31,43].

The detection occurs when the odorants penetrate into the nasal cavity and reach the olfactory mucosa. The odorants then interact with specific ORNs in the olfactory mucosa. Once an ORN is activated by an odorant, a nervous influx is sent to the cortex via the olfactory bulb. Readers interested in the mammalian olfactory epithelium and the perception of odor coding are invited to view an excellent review by Kurian and colleagues published in 2020 [44].

## 4. Viral Infection Causing Olfactory Dysfunction

The fact that the olfactory receptor neurons (ORNs) are found in the nasal cavity and expressed in the OE makes them directly exposed to all kinds of air-bound and air-way pathogens that make the ORNs vulnerable. Whether the cause is physiological or pathological, the lifespan of ORNs is relatively short with a few weeks in the OE. Moreover, stem cell reprogramming ensures the continuous regeneration of new ORNs from OE basal cells either in a physiological turnover of ORNs or in response to inflammation and OE severe damage mediated by neural injury [45,46,47]. Several airway pathogens, such as viruses, are causing damage to OE, particularly through the sustentacular cells, triggering anosmia, hyposmia, phantosmia, or parmosmia in mammals [7,9,42,48,49]. Many respiratory tract infections due to viruses like RV, PIV, RSV, CoV, and Epstein-Barr viruses (EBV) [4,8,9,10,11] have been involved in the development of olfactory disorders such as partial or total loss of smell. Doty and others have termed this pathology as virus-induced olfactory dysfunction (PVOD) [3,5,6,50]. Viral infection destroys many cells within the apical layer of the OE, which could lead to ORN functional impairment in the nose. Interestingly, the OE basal cells can constantly replace damaged ORNs with new olfactory neurons, allowing patients to recover functional olfactory responses [46,47]. In the following sections, the common viral infection of the URT leading to olfactory dysfunction like anosmia, hyposmia, phantosmia, and parmosmia [4] in animal models and humans will be discussed.

## 5. Viruses Impacting Respiratory System

The respiratory system is exposed to the environment and is in permanent contact with air-way pathogens like viruses. A recent investigation in humans has identified 18 viruses in patients with PVOD. Several known viruses are associated with olfactory impairment, and it is crucial to investigate the mechanisms of infection as well as the specific receptors each virus targets (Table 1).

### 5.1. Case of the Parainfluenza Viruses

The presence of parainfluenza type 3 (PIV3) was observed in human nasal epithelial cells (HNECs) from 88% of patients with PVOD as compared to 9% of control patients [51]. This data suggests the potential involvement of PIV3 infection in the upper airway pathology. PIV3 in the turbinate epithelial cells of PVOD is responsible for 60% of hyposmia and 40% of anosmia in patients [51]. PIV3 has been shown to infect the HNECs and, therefore, exacerbate the production of IFN-γ and pro-inflammatory cytokines [52]. This data is in line with a previous study suggesting that PIV3 may cause olfactory dysfunction through mechanisms other than nasal obstruction in patients [9].

### 5.2. Case of the Sendai Virus (SeV) and Possible Interaction with PIV

The SeV, the murine counterpart of the human PIV, has been shown to directly infect the mouse brain via the olfactory neurons [53]. Another investigation demonstrated that SeV infection led to impairing mouse olfaction. Interestingly, the virus persists in OE and OB tissues for over two months and reduces the regenerative power and functionality of the ORNs [54]. Very recent findings have shown that the depletion of nasal cilia via the silencing of CEP83, a protein critical for motile cilia formation in all ciliated cells, would impede the PIV infection. Indeed, the PIV receptor CX3CR1 colocalizes to motile cilia and is a plausible viral entry mechanism into the cells [55]. An additional study showed that the intercellular adhesion molecule-1 (ICAM-1) and related cytokine molecules are induced by PIV3, and it is thought that this activation participates in the inflammation during infection by viruses [56,57] (Figure 1). Further research using the power of the transcriptomic analysis is necessary to help delineate the role of the PIV and implicated mechanisms in the development of the broad range of olfactory dysfunctions in patients and to find host-response transcript signatures for possible treatments. The study of the entire RNA transcripts in a biological sample is referred to as transcriptomics, technically available as microarrays and RNA sequencing (RNA-seq) [58]. Only recently, some precision medicine trials using transcriptome analysis have been applied in the field of cancer [59,60,61,62,63]. Data from the study of Rodon and colleagues suggest that clinical trials using transcriptomics analysis can increase the number of patients matched to drugs [62]. Recent findings on the molecular basis of neuroimmune responses revealed that the transcriptomic results are in line with a few previously reported studies of respiratory viral infections like Influenza A virus, RV, and RSV. Moreover, these data highlighted putative biomarkers of interest as a direct reflection of each virus infection and deserve further investigation for the evaluation/prediction of future innovative treatments [64,65,66].

### 5.3. Case of the Respiratory Syncytial Virus (RSV)

Recent findings have demonstrated that the RSV can infect olfactory sensory neurons (OSNs) in the nasal cavity of the mouse before accessing the central nervous system of the animal [67]. As a PIV, the RSV targets the cilia of the epithelial cells in the airways by fusionning its F-glycoprotein to the cellular receptor human nucleolin (NCL). RSV also uses another mechanism that activates protein kinases like IGF1R to get into the cells [56,68,69]. Furthermore, recent studies have shown the essential role of the nucleolin RNA binding domain RBD1,2 in allowing the infection of RSV [70]. Previous works have pointed out the importance of ORN progenitors in the turnover. They showed that RSV infection causes SOX2+ ORN progenitors damage prior to the manifestation of ORN impairment. Unfortunately, the airway allergy seems to amplify this damage induced by the RSV infection, leading to a possible loss of OMP+ ORNs [13]. Transcriptomic analysis demonstrated that olfactory signaling is among the altered pathways in patients suffering from RSV infection. This finding further supports previous work that described RSV as a causative agent of post-viral olfactory dysfunction. Interestingly, the authors highlighted that this molecular signaling could be a promising future route to investigate drug targets against RSV infection [3,34]. Sourimant et al. have recently shown that 4′-fluorouridine, a ribonucleoside analog, inhibits RSV in a selective manner in cells and human airway epithelia organoids. Although this oral therapeutic drug is efficient in small animal models, several years of investigations are needed before predicting/evaluating the outcome of applying such a type of molecule in humans [71] (Figure 2). Furthermore, many efforts are underway in the phase of clinical trials when mRNA vaccines are combined with antigens to fight multiple respiratory viruses, including the RSV [72,73].

**Figure 2 microorganisms-12-00540-f002:**
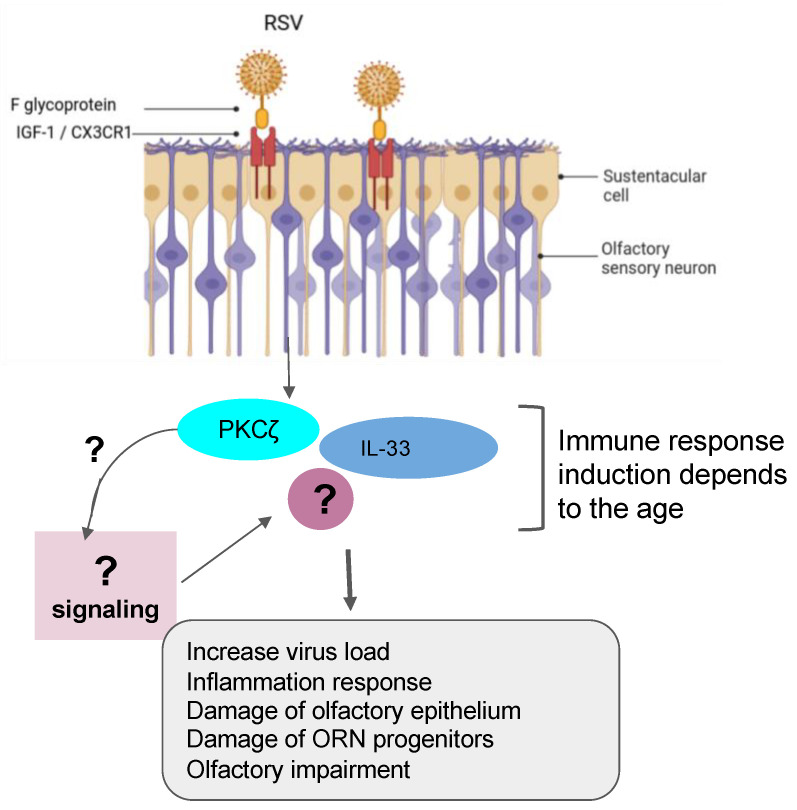
Role of RSV in olfactory impairment. RSV induced the inflammation response related to the age group with damage of olfactory epithelium and OSN progenitors and the impairment of olfactory response, leading to more virus susceptibility.

**Table 1 microorganisms-12-00540-t001:** Summary of virus-implication in PVOD.

Viruses	Animal or Cell Model	Effect of Respiratory Viruses on the Nasal Epithelial System	Comments	References
Parainfluenza virus	HNECs	PIV3 infection enhances the production of IFN-γ and generation of RANTES	IFN-γ protein was observed in non-infected cells for up to 72 h. 24 h post viral infection, were sufficient to notice significant increase of the IFN-γ but RANTES was detected only after 48 h. Study using epithelial primary cells or animal model to help decipher the role of IFN-γ in PIV3 induced PVOD. Data from Jun Tian et al. study provided some explanations on the later point.	[52]
	ALI-cultured NHE cells	Inhibition of cilia or microvilli may prevent airway entry and PIV spread throughout the viral-CX3CR1 receptor interactions in the ciliated HNEs	Presence of nasal cilia enhances PIV infection. Attenuation of virus trafficking might occur via CX3CR1 inhibition within the cilia as compared to TMPRSS2 inhibitor in SARS-CoV-2 propagation.	[55]
Sendai virus (SeV)	C57BL/6 mice and primary OSNs cultures	SeV prevents the primary cells from taking up the Ca^2+^ in the presence of odorants therefore altering directly the function of the OSN,	SeV is the murine counterpart of the human PIV. SeV triggers declination of the mouse olfactory function reaching a peak 15 days post-infection and persisting at least for 2 months in approximately 33% of animals The virus decreases apoptosis and cell proliferation. The normal cell turnover is lacking which may explain the inability of the olfactory epithelium to normally regenerate. The inflammation after a virus infection may also be considered as an additional mechanism in driving the olfactory neuron impairment.	[54]
	HT1080 cells	ICAM-1 is activated by PIV3 throughout the induction of the JAK/STAT signaling pathways.	The induction of ICAM-1 would trigger the inflammation during PIV3 infection. This induction is IFN signaling-independent.	[57]
Respiratory syncytial virus (RSV)	Mice and airway organoid cultures	Prefusion RSV-F glycoprotein is required to bind with the IGF-1 receptor. Such association triggers the activation of PKCζ.	Reduction of viral replication and pathology in RSV-infected mice after blunting PKCζ. Nucleolin-specific antibodies decrease RSV infection as well as RNAi designed to cellular nucleolin expression inhibition of RSV infections when a specific molecule targets the nucleolin RNA binding domain RBD1,2 of the virus. Preventing the linkage between RSV-F glycoprotein and IGF-1 to occur could form the basis of new therapeutics to treat RSV infection.	[68,69,70]
	Female BALB/c mice	Reduced SOX2^+^ ORN progenitors was enhanced and prolonged in allergic mice infected by RSV.	Expression of SOX2^+^ ORN progenitors were affected in the nasal mucosa due to RSV transient infection. Delayed RSV clearance and exacerbated progenitors damage observed in airway allergy mice.	[13]
Rhinoviruses	Sinus epithelial tissue, HeLa R-19, mouse L cells	The following three major types of cellular membrane glycoproteins ICAM-1, LDLR and CDHR3 are targeted by RV to gain entry into the host cell.	Enhanced expression of these receptors increased RV interaction and enabled replication in nasal host cells. Reduced virus adhesion throughout receptor-antibody would neutralize RV entry and attenuate cell inflammation leading to olfactory dysfunction.	[74,75,76,77]
	Sinonasal epithelial cells	H_2_O_2_ significantly reduced the production of (IFN-β) and type III (IFN- λ1 and λ2) interferons that was upregulated in cells infected with RV.	Decrease expression of TLR3, RIG-1, MDA5, and IRF3 was observed in cells pretreated with H_2_O_2_. Nrf2 siRNA showed decreased secretion of anti-viral interferons in transfected cells. Oxidative stress inhibiting the anti-viral interferons in the sinonasal mucosa due to RV infection is a potential road to prevent inflammation-induced PVOD.	[78]
SARS-CoV-2	ciliated HNE cells	SARS-CoV-2 establishes a link with angiotensin-converting enzyme 2 (ACE2) receptor within the airway multicilia to traverse the mucus- mucin protective barrier.	Depleting cilia prevents virus virus entry. Inhibition of PAK1, PAK4 and SLK kinases activity reduces virus spread in mice. Depleting cilia in nasal epithelial cells does not affect ACE2 and transmembrane serine protease 2 (TMPRSS2 levels). More insight into cilia and microvilli reprogrammation for virus entry would identify marker candidates for treatment to block airway replication of SARS-CoV-2.	[55]
	K18-hACE2 mice, golden Syrian hamsters, cellular models	SARS-CoV-2 targets the sustentacular and Bowman gland cells by binding to their ACE2 and TMPRSS2 proteins.	Exponential growth of the virus leading to the destruction of these supporting cells. Significant reduced thickness of the mucus layer could be related to the Bowman’s glands deterioration which are the precursor of the mucus. Healthy mucus is crucial for odor detection as it enables odorants to diffuse to olfactory receptors. Retraction of OSN cilia, although the mature OSN are free of SARS-CoV-2 viral load. SARS-CoV-2 alterates throughout the supporting cells not only the structure of mucus but also the OSN cilia which contribute to the olfactory dysfunction.	[79,80,81,82,83,84,85,86]
	K18-hACE2 mice, golden Syrian hamsters, cellular models	Supporting cells infected by SARS-CoV-2 affect the expression of the GLUT1/GLUT3 that interrupt the glucose trafficking to the cilia of the ORN in the mucus.	Sustentacular cells and Bowman gland cells supply additional glucose to the cilia that is necessary for the ORN to respond to odorants. SARS-CoV-2 uses the internal glucose as a fuel to maximize its replication thus preventing the cilia to undergo odorant-induced response of the ORN. Loss of glucose normally supplied by sustentacular cells and Bowman gland cells. SARS-CoV-2 exhibits supporting cell damage that consequently trigger the interruption of the supply of additional glucose to the cilia of the ORN via GLUT1/GLUT3.	[87,88,89,90,91]
	ALI and iALI cells	SARS-CoV-2 activates key molecular mechanisms after infecting ALI and iALI cellular models.	SARS-CoV-2 infection in ALI and iALI cells activates inflammatory state and innate immune response. The virus induces epithelial disruption, loss of mature ciliated cells. miRNAs like MIR138 regulates ISG15 expression and plays a role in the virus-induced pro-inflammatory genes activation. The option of miRNAs is promising for COVID-19 treatment and/or prevention.	[92]

Recently, RVs were shown to be among the more predominant causative agents of PVOD in patients. The study showed that patients with anosmia were higher than those with hyposmia (58.8% vs. 19.0%, *p* = 0.018) [8]. In line with these findings, it is tempting to suggest that the persistence of the virus could be one factor that governs a more severe injury to the olfactory system. In fact, RVs primarily invade the ciliated respiratory epithelial cells via the glycoprotein members such as the intercellular adhesion molecule 1 (ICAM-1), the low-density lipoprotein receptor (LDLR) family members, and the cadherin-related family member 3 (CDHR3) [74,75,76,77]. RV infection induces Toll-like receptor 7 (TLR7) and retinoic acid-inducible gene I (RIG-1) that trigger the activation of cytokine expression (type I and type III IFNs) [93,94,95]. Understanding the mechanisms of RV viral-induced asthma for new therapeutic directions has gained more attention in the recent past [96,97,98]. Papi et al. have investigated the role of reducing agents such as DMSO on RV infection in the nasal epithelium. They showed that rhinovirus-induced ICAM-1 mRNA expression was inhibited by reducing agents in a dose-dependent manner. Interestingly, NF-κB and TNF-α activation, which is necessary for the ICAM-1 promoter, was completely abolished in those treated epithelial cells [99]. Moreover, it has been shown that CDHR3 genetic variants impact on the severity of RV-related pediatric respiratory tract infections by upregulating the epithelial expression of RV receptors, thus helping clinicians predict the susceptibility and severity of RV infection [100]. Complementary recent studies have demonstrated that both vitamin D and hydrogen peroxide play a critical role in attenuating the RV, mediating the ICAM-1 activation and the production of type I (IFN-β) and type III (IFN- λ1 and λ2) interferons respectively [78,101] (Figure 3). To further support these collective findings, it would be interesting to document transcriptomic profiles from animal and cellular models infected by RVs and treated by those molecules. These works would shed light on the importance of genes, including the oxidant biomarkers to be considered for the future in the development of the treatment against RV infection.

## 6. Mechanisms of SARS-CoV-2 Mediating the Loss of Smell

The post-COVID and the long-term-COVID have both tremendously triggered a lot of complications in different human systems. The loss or reduction of smell, among other complications of the nervous system, is an associated symptom for patients affected by different variants of COVID-19, including the omicron variant [102,103,104,105,106]. Moreover, studies reported that the prevalence of olfactory dysfunction differs greatly between populations and approaches [106,107,108]. Currently, many COVID-19 vaccines are authorized to help protect and eliminate the virus. The COVID-19 pathology and the cellular mechanism by which the olfactory dysfunction occurs have gained a lot of attention since the pandemic, and researchers are still investigating underlying signaling and complications (Table 1) [55,91,92,106,109,110,111,112]. Earlier in the pandemic, reports hypothesized that five potential mechanisms were considered to get insights into the olfactory dysfunction in COVID-19 patients: (1) obstruction/congestion and rhinorrhea of the nasal airway, (2) damage and loss of ORNs, (3) Olfactory center damage in the brain, (4) damage of the olfactory supporting cells in the OE, and (5) Inflammation-related olfactory epithelium dysfunction [107,113]. Butowt et al. have recently reviewed that at least the following hypotheses (1)–(3) turned out to be implausible in explaining the olfactory dysfunction in patients [113]. This allegation is further confirmed by very recent studies showing that SARS-CoV-2 infection significantly increased the expression of interferon-stimulated and inflammatory genes. Alteration of extracellular matrix genes was also observed in ALI and iALI-infected cells [92]. Here, we will particularly review the mechanisms related to the second, the fourth, and the fifth scenarios according to the available findings. Healthy sensory cilia of ORNs in the olfactory epithelium are crucial in perceiving odorant molecules before sending the information to the olfactory bulbs and then to the upper parts of the brain [45]. It has been reported in humans that SARS-CoV-2 may indirectly affect the olfactory cilia, hindering the smelling system’s efficacy [114]. Reports suggested that ORNs lack the ability to express the entry proteins of SARS-CoV-2 in the OE. The virus seems to establish first contact in human nasal epithelia by binding its spike S protein to specific cells in the OE [115]. These reports are confirmed by a study based on in silico data, predicting that mature ORNs do not express the virus entry protein, the angiotensin-converting enzyme 2 (ACE2), and therefore are not likely to be infected by SARS-CoV-2 [81]. Furthermore, supporting data by Bryche et al. showed that SARS-CoV-2 was not detected in the ORNs of golden Siryan hamsters [85]. However, in a few cases, authors suggested that SARS-CoV-2 could infect ORNs in hamsters [111]. Based on the fact that COVID-19-related loss of smell disappeared within 1–2 weeks, while the regeneration of dead ORNs needs more than two weeks, many data tend to conclude that COVID-19-related olfactory dysfunction (OD) is not directly associated with the impairment of the ORNs [107,113,115,116,117]. Consequently, studying the entry protein expression within the cells in the OE will help to understand the sensitivity of the OE to SARS-CoV-2 infection related to the high prevalence of ODs in patients. Many groups are now interested in the organization of the sustentacular cells in the OE and thought that they might play a central role in leading to OD. A high level of expression of ACE2 and the transmembrane serine protease 2 (TMPRSS2) is particularly found on the sustentacular cells, suggesting a path to the neurotropism of SARS-CoV-2 in the OE. The ACE2 and TMPRSS2 are respectively known as the SARS-CoV-2 receptor and the SARS-CoV-2 cell entry-priming protease. ACE2 is found mainly on different parts of the sustentacular cells, both in humans and mice. The ACE2 and TMPRSS2 genes tend to be co-regulated [113,118,119,120,121,122]. Different approaches using tissues, cells, and organ systems in humans, golden Syrian hamsters, and hACE2 transgenic mice have been employed to study the pathological impact of the SARS-CoV-2. Here, we discussed findings related particularly to the OE in inducing ODs in humans. The spike protein (S protein) of SARS-CoV-2 mediates the passage of the virus into the host cell by fusing the viral and host cell membranes. In fact, via his spike S, SARS-CoV-2 employs ACE2 as the host functional receptor and TMPRSS2 as the cellular priming protease facilitating viral uptake, both signalings being confirmed by Single-cell RNA sequencing (scRNA-seq) datasets from the Human Cell Atlas consortium [123,124,125]. Another study showed that SARS-CoV-2 Nucleocapsid protein (NP) was observed in human OE through the neuronal marker Tuj1 9 h post-infection. This data further supported the enrichment of ACE2 in human olfactory sustentacular cells [119,126]. Earlier in the pandemic, the golden Syrian hamster was used as a model to document the pathology of SARS-CoV-2 in the OE post-infection. Reports showed that the sustentacular cells are rapidly infected by SARS-CoV-2. This viral neurotropism is associated with a massive recruitment of immune cells in the OE and lamina propria, which could drive the disorganization of the OE structure [85]. This study is consistent with a high level of Tumor Necrosis Factor α (TNF α) observed in OE samples from COVID-19-suffering patients and in ALI and iALI-infected cells [79,92]. Furthermore, the inflammation induced by SARS-CoV-2 infected supporting cells may play an important role in the onset and persistence of loss of smell in patients. This SARS-CoV-2-associated inflammation status was confirmed by the transcriptome of the in vitro human airway epithelium and by analyzing the expression of selected targets in the olfactory bulb using RNA-seq and RT-qPCR tools. Interestingly, this study showed that the proinflammatory markers, including NFKBIA, CSF1, FOSL1, Cxcl10, Il-1β, Ccl5, and Irf7 overexpression continued up to 14 dpi when animals had recovered from ageusia/anosmia [92,127]. These findings are in line with a very recent study showing the implication of immune cell infiltration and altered gene expression in OE in driving persistent smell loss in a subset of patients with SARS-CoV-2. Moreover, this study particularly demonstrates that T cell-mediated inflammation lasts longer in the OE after the acute SARS-CoV-2 infection has been eliminated from the tissue, suggesting a mechanistic insight into the long-term post-COVID-19 smell loss [105]. The OE disorganization is followed by a drastic deterioration of the cilia layer of the ORNs that leads to the impairment of the olfactory capacity of the animal [85]. Investigations in humans and hamsters using, respectively, Transmission Electron Microscopy (TEM) studies and Scanning Electron Microscopy (SEM) analysis showed various levels of cilia height that undergo regeneration in the course of patient recovery, including smell restoration. Data using the golden Syrian hamster showed that the regenerated cilia in the epithelium are accompanied by a decreased expression of FOXJ1+, highlighting the importance of this marker in respiratory ciliogenesis. This later finding by Schreiner et al., could in part shed light on the inquiry of how could we regenerate cilia during patient recovery, although a lot needs to be done in the roadmap of treating loss of smell related to nasal cilia deterioration by SARS-CoV-2 [55,104,128,129] (Figure 4; Table 1). 

According to the literature, different variants of SARS-CoV-2 do not directly target the ORNs in the OE; instead, they are found in the majority expressed in the sustentacular cells [42]. A recent study by Seehusen et al. showed that K18-hACE2 transgenic mouse expressing the human ACE2 is highly sensitive to at least five variants of SARS-CoV-2 that infected not only the supportive cells in OE and the respiratory epithelium but invaded the CNS of the animal five days post-infection. Interestingly, the expression of hACE2 seems to convey higher binding affinity when compared to the wild-type mouse [130]. Using this transgenic mouse is revealed to be a serious option for therapy development against loss of smell as these animals exhibited low mortality when treated with COVID-19 convalescent antisera [80,130]. For instance, the miRNA is shown to play a crucial role in the regulation of immune genes deregulated, and the development of miRNA antagonists or mimics seems to be a promising new therapeutic strategy for the treatment of patients with COVID-19 on other respiratory viruses-induced PVOD [72,73,92,131,132,133].

It is now accepted that ACE2 is not the only obligate entry for SARS-CoV-2 as it has been suggested that molecules including PIKfyve or neuropilin-1 (NRP-1) may participate in SARS-CoV-2 entry [82,83,84]. Like ACE2, NRP-1 is highly expressed in the respiratory and olfactory epithelium, which further supports the infectivity and entry of SARS-CoV-2 in the human OE. NRP-1 is not only found in supportive cells but is expressed in nearly every cell type in the nasal passages, including the ORN, therefore giving SARS-CoV-2 a route to access those cells and impair the olfactory response. Interestingly, Daly et al. demonstrated that the selective inhibition of the S1-NRP-1 interaction reduces SARS-CoV-2 infection [84,86,87].

Taken together, the high expression of ACE2, TMPRSS2, and NRP-1 in supportive and other olfactory cells and their impact on olfactory neurophysiology maintenance and in the development of human olfactory pathophysiology supports them as potential targets for signaling-based therapeutics of olfactory dysfunction.

## 7. Pathological Implications of ARV Co-Infections

Coinfections with SARS CoV-2 and other ARVs may have tremendous pathological implications with potentially fatal outcomes [134,135,136,137]. The proportion of coinfections in SARS-CoV-2-infected patients and the types of viruses involved vary in different regions of the world depending on the sensitivity of the diagnostic tests used, population studied, climate, sampling period, and temporal variations in viral epidemiology. Some studies report proportions that vary from 1.7 to 20% [134,138,139,140,141,142,143]. Authors hypothesize that competitive advantage may play a role in SARS-CoV-2 interaction with other respiratory viruses during coinfection, and this is one reason why the coinfection rate in SARS-CoV-2 patients is much lower [142]. In many cases, the detection of RSV, influenza A, and other coinfections led to changes in clinical management in admitted patients to the medical intensive care unit [136,137].

Picornaviruses, influenza A and B, RSV, and parainfluenza are among the most detected viruses in COPD exacerbations [135]. Increased susceptibility to viral respiratory infections such as SARS-CoV-2 has been reported in chronic obstructive pulmonary disease (COPD), often worsened by bacterial co-infections and leading to serious clinical outcomes [137]. 

Trifonova et al. reported a more intensive replication of SARS CoV-2 and influenza viruses compared to that of other respiratory viruses involved in coinfections. The level of the viral load involved in mixed infections depends largely on the time of exposure to one virus relative to the other. These authors explain their findings by the fact of positive or negative interactions between SARS-CoV-2 and other respiratory viruses via interferon-mediated or other immunological mechanisms [134].

## 8. Clinical Trials for Respiratory Virus Infection

The protection against viral infection affecting human respiratory tracts constitutes a huge challenge and, therefore, forces clinicians and scientists to develop effective antivirals or vaccines to alleviate the death rate in patients [144]. Although mRNA COVID-19 vaccines are being authorized and administrated since 2020, for most of the respiratory viruses, there is no vaccine currently available. Moreover, in general, vaccination is efficient before the onset of the viral season infection. For example, the RSV seasonality was affected by the COVID-19 pandemic [145]. Currently, two molecules are being used against RSV: ribavirin for treatment and palivizumab for prevention. Many other antivirals to prevent RSV are in experimental stages as well as the use of anti-inflammatory drugs is to be considered, although the efficiency is to be verified [146]. To date, several recombinant RSV subunit vaccines are in different clinical phases 1 and 2 trials [147,148], which is in line with the undergoing clinical development of new vaccines for protecting the children and the elderly against RSV infection [149,150,151,152,153]. Very recently, the Food and Drug Administration (FDA) has approved the first vaccines for the prevention of RSV-associated low respiratory tract (LRT) in adults 60 years old and over. Both the GSK RSV vaccine and the Pfizer RSV vaccine were evaluated for their efficacy and safety. The efficacy of 1 dose of the GSK and Pfizer vaccine in preventing medically attended RSV-associated LRT was 77.5% and 81.0%, respectively, and both were safe according to the low severe reactogenicity events (group participants vs. control group participants) [145]. For instance, it is to be elucidated whether those new RSV vaccines will help to prevent the loss of smell within the URT in adults. The inhalation of ALX-0171 to prevent RSV infection in infants and toddlers is at Phase IIb clinical trial and is still under investigation for approval by the US FDA [154]. Regarding the PIV, the treatment is still for symptoms, although several investigations are being conducted to prepare a viable vaccine [155,156]. A study in 2019 has shown that DAS181, a sialidase fusion protein, may have clinical activity, particularly in immunocompromised patients with PIV, based on the data from the clinical phase 3 trials [157]. Karron and colleagues have recently conducted a phase I clinical trial of the live-attenuated recombinant human PIV2 in adults, as well as in children and seronegative children. In fact, rHPIV2-15C/948L/∆1724 was appropriately restricted in replication in adults and HPIV2-seropositive children but was overattenuated for HPIV2-seronegative children. Their evaluation showed that the rHPIV2-15C/948L/∆1724 represents less attenuated alternatives for pediatric vaccine development [158]. Further testing and clinical trials are required in the fight against the human PIV, especially PIV type 3, as it is considered to be the most virulent form of human PIV [159]. To our knowledge, there are no available vaccines or drugs against rhinovirus infections. Physical distancing seems to be not effective in reducing the transmission of this virus. But, it is essential to notice that rhinovirus infection is not associated with a significant rate of hospitalization or death [160]. Several researchers have demonstrated that daily dosing of 1 million IU of intranasal IFN-alpha gave 75–87% protection against rhinovirus infection [161,162]. The use of zinc gluconate lozenges has proven to be an effective treatment in reducing symptoms of rhinovirus infection [163]. For instance, if the vaccine against rhinovirus is not available, one may consider taking an antiviral with other antimediators for better protection from infection by the rhinovirus [164]. According to the WHO, COVID-19 treatment guidelines are evolving and fortunately several COVID-19 vaccines have been approved to actively immunize the general population. One can track the COVID-19 vaccines following this documentation [152].

## 9. Conclusions and Perspective

Our literature review further confirms the previous extended investigations showing that loss of smell and taste are among the key associated symptoms with most COVID-19 variants, including the omicron variant, which causes runny nose, headache, fatigue, sneezing, and sore throat [165]. The last three years have been an important rush towards deciphering the underlying mechanisms the SARS-CoV-2 deploys to impair the olfaction in infected patients. Furthermore, it is interesting to delineate the similarities and differences between the molecular mechanisms of SARS-CoV-2 and the other respiratory viruses induced olfactory dysfunction. Both SARS-CoV-2 and non-SARS-CoV-2 attach to the cilia during the initial stages of infection to later enter the nasal epithelium [55]. Interestingly, all these findings underline the importance of the immune-mediated inflammatory injury to the olfactory neuroepithelium that is now accepted as a consequence of those URT virus infections. However, the molecular signature related to the type of olfactory dysfunction, according to Table 1, seems particular for each respiratory virus. The animal or cellular models being used and the seasonal periods could add more complexity to the putative mechanisms for viral infection-induced olfactory dysfunction. Investigative literature on the COVID-19 mechanistic route has made clear that this virus seems to attach to ACE2-TMPRSS2 complex and/or NRP-1 on the host cell prior to infection and later triggers intrinsic immune responses. In the nasal mucosal microenvironment, those markers play a crucial role in inflammatory response mechanisms and are confirmed by several recent studies on understanding SARS-CoV-2 invasion [80,82,83,84,86,130,166]. Currently, the mechanism of how SARS-CoV-2 causes smell loss is widely documented, and more investigations are needed on the non-SARS-CoV-2 to complete the picture of comparing the particularity of each respiratory virus causing URT-related PVOD. For instance, this work emphasizes the urgency and necessity of finding an adequate therapeutic solution against COVID-19 and other respiratory viral pathogens-induced olfactory dysfunction. In addition, the mechanisms of taste dysfunction due to COVID-19 infection are not discussed in this review. But, it would be interesting to decipher the possible pathogenesis between ageusia and anosmia and other types of PVOD in COVID-19 and other non-COVID-19 patients in the future. 

## Figures and Tables

**Figure 1 microorganisms-12-00540-f001:**
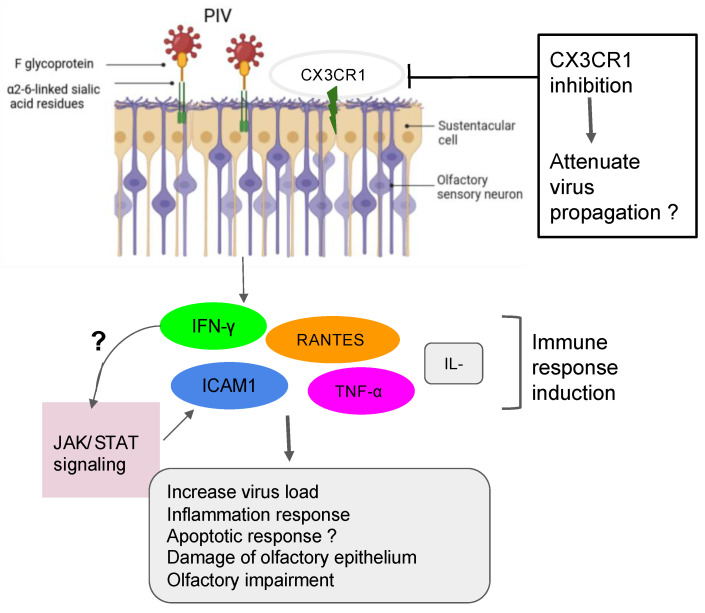
Action of PIV in the olfactory impairment. PIV induced the inflammation response with damage to olfactory epithelium and the impairment of olfactory response, leading to more virus susceptibility.

**Figure 3 microorganisms-12-00540-f003:**
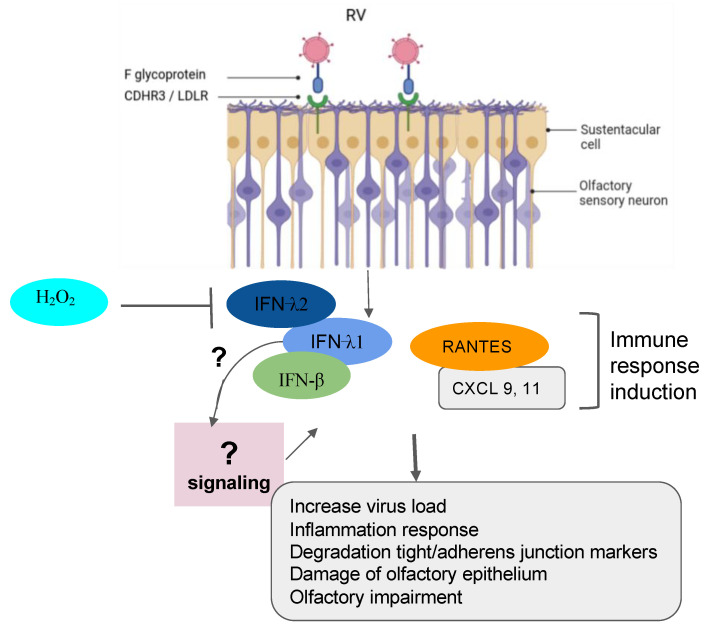
Implication of RV in olfactory impairment. RV induced the inflammation response with damage to the olfactory epithelium, the degradation of the tight junction and adherent junction markers, and the impairment of olfactory response leading to more virus susceptibility.

**Figure 4 microorganisms-12-00540-f004:**
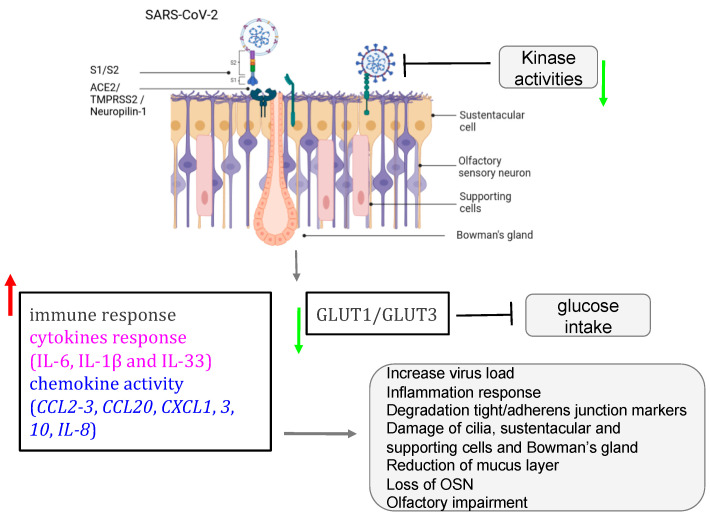
Role of SARS-CoV-2 in olfactory impairment. SARS-CoV-2 induced disruption of the nasal epithelium, with loss/damage of olfactory sensory neurons, sustentacular cells, Bowman’s gland, and supporting cells. All figures were created with BioRender.com (accessed on 7 October 2023).

## Data Availability

Data are contained within the article.

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
