# Peer review of "SARS-CoV-2 and Other Respiratory Viruses in Human Olfactory Pathophysiology"

_microorganisms, 2024, doi:10.3390/microorganisms12030540_

Round 1

Reviewer 1 Report

Comments and Suggestions for Authors

Check the use of acronyms; if they were already defined at the beginning, entering their whole meaning is no longer necessary.

I suggest putting subtitles for each virus in section 5 (Viruses impacting the Respiratory System)

In Table 1, I suggest putting "some" animal or cell model.

Figure 1 should be separated into four figures with a caption for each one. Additionally, it requires higher resolution.

Line 214, "et al."

Line 255 in silico

I suggest a section on the pathological implications of ARV co-infections.

Author Response

We want to thank the reviewers for their constructive comments on our manuscript. You will find in this letter a point-by-point response to the reviewers comments concerning our manuscript entitled « SARS-CoV-2 and other respiratory viruses in human olfactory pathophysiology ».

Reviewer 2 Report

Comments and Suggestions for Authors

During the past few yrs it has become evident that infections with respiratory viruses such as SARS-CoV-2 cause other debilitating diseases.  One unexpected finding has been that nearly 20-40% of the infected individuals end up with extra respiratory sequela which can include loss of taste and smell sometimes for many months.   Interestingly, for these chronic aftermath events, there is little to no correlation with the number of vaccinations or infections.  This area of research has been neglected before because the number of individuals impacted was lower and the intensity of the chronic sequelae was not as high as observed with SARS-CoV-2 infections. In this timely mini-review “SARS-CoV-2 and other respiratory viruses in human olfactory pathophysiology,” Serigne Fallou Wade and colleagues have attempted to review the literature on the consequences of upper viral respiratory infection and olfactory dysfunction.

- The translational aspect of the review is weak and needs substantial improvement.  The authors neglected to mention a few therapeutic modalities which have been tested in clinical trials against long-term effects of respiratory viruses. The authors must research clinicaltrial. gov, press releases from various sources, and other governmental or academic sources to put a paragraph or two about some of the modalities that are being tested in human trials.

1-      Lines 153-156, The authors stated that: “Further research using the powerful of the transcriptomic analysis is necessary to help to delineate the role of PIV and implicated mechanisms in the development of the broad range of olfactory dysfunctions in patients and to find host-response transcript signature for possible treatments.”   It is not clear how the transcriptomics can lead to treatments.  The authors should expand on this with a sentence or two.  

2-      Lines 174-177, the authors mention “ As this orally therapeutic drug is efficient in small animal models,  further investigations are needed to apply this molecule in human with the same yield [69] (Fig. 1B). Efforts are underway in the phase of clinical trials when mRNA vaccines are combined to antigens to fight multiple respiratory viruses including SRV [70, 71].” Just because a chemical works in small animals the authors must refrain from jumping to clinical studies. Many yrs of R&D separate experimental work from clinical studies.  The authors must reconsider how they would like to expand on these findings

3-      Throughout the review, the authors mention how something or a mechanism has altered the viral infection and then they postulate using the chemical or the MOA in humans as a treatment.  For example lines 214-226 the authors mention “Papi et al, have investigated the role of reducing agents such DMSO  on RV infection in the nasal epithelium. They showed that rhinovirus-induced ICAM-1  mRNA expression was inhibited by reducing agents in a dose-dependent manner. Inter-  estingly, NF-κB and TNF-α activation which is necessary for the ICAM-1 promoter, was  completely abolished in those treated epithelial cells [99]. Moreover, it has been shown  that CDHR3 genetic variants impact on severity of RV-related pediatric respiratory tract infections by upregulating the epithelial expression of RV receptors thus helping clinicians to predict the susceptibility and severity of RV infection [100]. Complementary recent studies have demonstrated that both vitamin D and hydrogen peroxide play a critical role in attenuating the RV mediating the ICAM-1 activation and the production of type I (IFN-β) and type III (IFN- λ1 and λ2) interferons respectively [77, 101] (Fig. 1C). The identified mechanisms focusing on oxidant biomarkers could be a plausible road for reducing the clinical severity or treating RV infection induced-olfactory dysfunction in patients.”    For example just because bleach kills SARS-CoV-2, we do not use it to treat humans with bleach. Using reducing agents in vitro is very different than applying the concept to clinical trials and so on.  The authors must control their enthusiasm for these types of temptations and be realistic of what can be applied to human clinical trials.

4-      There is little to no correlation between using antivirals such as Paxlovid and reduction in chronic pathologies such as loss of taste and smell.  This is important to note throughout the review and the review should be guided by high-quality publications.

5-      The reviewer was disappointed that the role of inflammation in combination with antiviral/or viral infection was not fully reviewed here.

6-      A lot of the review was conducted on early research and little space was used for clinical studies or how to approach clinical studies for this massive problem.  The authors must rethink how they have approached the mini review and consider spending a few paragraphs on clinical studies which may have implications for this mini-review.

Comments on the Quality of English Language

Need moderate revision for the use of English language

Author Response

(The authors gave the same response as above.)

Round 2

Reviewer 2 Report

Comments and Suggestions for Authors

No additional comments

Comments on the Quality of English Language

Some editing is required